

# KSFinder—a knowledge graph model for link prediction of novel phosphorylated substrates of kinases

Manju Anandakrishnan[1], Karen E. Ross[2], Chuming Chen[1], Vijay Shanker[1], Julie Cowart[1] and Cathy H. Wu[1,2]

[1] Center for Bioinformatics and Computational Biology, University of Delware, Newark, DE, United States of America
[2] Department of Biochemistry and Molecular & Cellular Biology, Georgetown University Medical Center, Washington, DC, United States of America

Corresponding author
Manju Anandakrishnan, manjua@udel.edu

## ABSTRACT

**Background**. Aberrant protein kinase regulation leading to abnormal substrate phosphorylation is associated with several human diseases. Despite the promise of therapies targeting kinases, many human kinases remain understudied. Most existing computational tools predicting phosphorylation cover less than 50% of known human kinases. They utilize local feature selection based on protein sequences, motifs, domains, structures, and/or functions, and do not consider the heterogeneous relationships of the proteins. In this work, we present KSFinder, a tool that predicts kinase-substrate links by capturing the inherent association of proteins in a network comprising 85% of the known human kinases. We also postulate the potential role of two understudied kinases based on their substrate predictions from KSFinder.

**Methods**. KSFinder learns the semantic relationships in a phosphoproteome knowledge graph using a knowledge graph embedding algorithm and represents the nodes in low-dimensional vectors. A multilayer perceptron (MLP) classifier is trained to discern kinase-substrate links using the embedded vectors. KSFinder uses a strategic negative generation approach that eliminates biases in entity representation and combines data from experimentally validated non-interacting protein pairs, proteins from different subcellular locations, and random sampling. We assess KSFinder's generalization capability on four different datasets and compare its performance with other state-of-the-art prediction models. We employ KSFinder to predict substrates of 68 "dark" kinases considered understudied by the Illuminating the Druggable Genome program and use our text-mining tool, RLIMS-P along with manual curation, to search for literature evidence for the predictions. In a case study, we performed functional enrichment analysis for two dark kinases - HIPK3 and CAMKK1 using their predicted substrates.

**Results**. KSFinder shows improved performance over other kinase-substrate prediction models and generalized prediction ability on different datasets. We identified literature evidence for 17 novel predictions involving an understudied kinase. All of these 17 predictions had a probability score $\geq 0.7$ (nine at $> 0.9$, six at 0.8–0.9, and two at 0.7–0.8). The evaluation of 93,593 negative predictions (probability $\leq 0.3$) identified four false negatives. The top enriched biological processes of HIPK3 substrates relate to the regulation of extracellular matrix and epigenetic gene expression, while CAMKK1 substrates include lipid storage regulation and glucose homeostasis.
**Conclusions**. KSFinder outperforms the current kinase-substrate prediction tools with higher kinase coverage. The strategically developed negatives provide a superior generalization ability for KSFinder. We predicted substrates of 432 kinases, 68 of which are understudied, and hypothesized the potential functions of two dark kinases using their predicted substrates.

# INTRODUCTION

Phosphorylation, a reversible post-translational modification (PTM), is a common cell mechanism that regulates many cellular processes. Protein kinases are enzymes that catalyze phosphorylation by transferring a phosphate molecule to the substrate, thereby regulating the substrate protein's function (*Manning et al., 2002*; *Ardito et al., 2017*). Abnormal kinase regulation results in dysfunction of the substrate, leading to several known human disease conditions (*Fabbro, Cowan-Jacob & Moebitz, 2015*). These include but are not limited to diabetes, cardiovascular conditions, Alzheimer's, and different cancer types. Kinases have thus become a major class of potential therapeutic targets. Despite their important roles, many human kinases remain understudied and poorly understood (*Vlahos, McDowell & Clerk, 2003*). The Illuminating the Druggable Genome program (IDG) launched by NIH Common Fund lists 134 kinases as understudied druggable proteins (*The National Institutes of Health, 2023*). To advance the use of kinases in drug discovery, a better understanding of their phosphorylation network is essential (*Cohen, Cross & Jänne, 2021*).

## Current approaches

The experimental approaches that are widely used for studying phosphorylation (*e.g.*, mass spectrometry-based high throughput profiling) provide site-specific phosphorylation information; however, the identification of the involved kinases is challenging (*Engholm-Keller & Larsen, 2013*; *Xue & Tao, 2013*). There are experimental methods that provide kinase-specific information, such as kinase activity assays, but they are laborious, expensive, and time-consuming (*Cohen, Cross & Jänne, 2021*). Many approaches are also prone to false predictions due to the *in vitro* nature of the experiments (*Xue & Tao, 2013*). Computational tools offer an alternative that can predict the potential substrates for kinases and limit the pool of candidates for experimental testing. Several computational tools for predicting kinase-substrate relationships have been developed in the last couple of decades, but most of them cover less than half of the known human kinases, and they generally consider the interaction between the two proteins in a local space, such as linear motif-based, structure-based, sequence-based, and function-based (*Blom et al., 2004*; *Obenauer, Cantley & Yaffe, 2003*; *Xue et al., 2008*; *Horn et al., 2014*; *Song et al., 2017*).

### Knowledge graph embedding (KGE)

In this study, we used knowledge graph embedding to capture the latent association of kinases and substrates with other biological entities based on their heterogeneous relations including functions, biological processes, pathway associations, and the hierarchical information from Gene Ontology (*Ashburner et al., 2000*; *Gene Ontology Consortium, 2021*) and Protein Ontology (*Natale et al., 2011*). Currently, there are two KGE methods, LinkPhinder (*Nováček et al., 2020*) and PredKinKG (*Gavali et al., 2022*), that study phosphorylation and they have demonstrated superior performance over other non-graph-based prediction tools. Knowledge graph representation learning captures the latent patterns in the heterogeneous network that are generally overlooked by context-based feature selection procedures. Additionally, the high-level information used in the graph allows the prediction of direct and indirect phosphorylation links.

### LinkPhinder & PredKinKG

LinkPhinder uses protein motifs, kinases, and site-specific information as input to its knowledge graph and makes predictions using graph embedding (*Nováček et al., 2020*). In PredKinKG, *Gavali et al. (2022)* integrated the phosphorylation interactions at the protein level, along with functional annotations of proteins, and the hierarchical relationship of the annotated terms and proteoforms to predict kinase-substrate interaction. The method employed a two-step approach combining graph embedding with downstream classification and demonstrated superior performance over LinkPhinder. PredKinKG uses TripleWalk, a directed random walk algorithm coupled with the Continuous Bag of Words (CBOW) model for graph embedding, and a random forest machine learning algorithm for classification (*Gavali et al., 2022*). Though PredKinKG performs its embedding using kinase-substrate phosphorylation relations, it predicts the probability of the more general event of two proteins interacting with each other. Moreover, PredKinKG has an unbalanced representation of the entities in its negative and positive datasets for the classifier model, causing unintended bias in the supervised learning process. For instance, the substrate protein, gelsolin, occurs in 216 triples in the negative set whereas there are only two records involving the substrate in the positive set. As the negatives are generated computationally by different strategies, the biases in the entity representation influence the predicted outcomes. While this bias may not affect the knowledge graph embedding learning because synthetic negatives are generated randomly *via* the closed-world assumption (CWA) (*Nickel et al., 2016*), it influences the supervised classification model that is trained with the vectors of the kinases and substrates.

### Substrate specificities of serine/threonine kinome

*Johnson et al. (2023)* employed positional scanning peptide array (PSPA) and profiled the substrate specificities for 303 human serine/threonine kinases. Using position-specific scanning matrix and kinome wide dataset, they computationally annotated and ranked favorable kinases for 89,784 sites on different substrates. In addition to comparing KSFinder's performance with LinkPhinder and PredKinKG which report improved performance over context-based feature selection models, we compare KSFinder's

performance with the atlas of kinase-substrate specificity reported by *Johnson et al. (2023)* in their recently published work. We will refer to this atlas throughout the rest of the article as the 'Ser-Thr-KS atlas'.

### KSFinder

In this article, we present KSFinder, a model integrating knowledge graph embedding (KGE) and a multilayer perceptron (MLP) neural network for uncovering novel kinase-substrate relationships. Our model differs from PredKinKG by predicting the probability of phosphorylation rather than interaction and also addresses the bias in the negative dataset using a combinatorial approach for the negative generation. KSFinder uses ComplEx KGE algorithm, and a more sophisticated approach for downstream classification using the multilayer perceptron (MLP) neural network. The classifier was trained on the embedded vectors of the kinases and substrates extracted from the KGE and learned to discern true kinase-substrate links in the knowledge graph. We demonstrate KSFinder's performance by evaluating it on four datasets and by comparing it with other prediction models, LinkPhinder, PredKinKG, and Ser-Thr-KS atlas.

### Functional analysis

We mine the literature for evidence of novel predictions from KSFinder and assess the model's ability to correctly predict kinase-substrate pairs. In a case study, we extracted the high-confidence substrates for two understudied kinases, HIPK3 and CAMKK1, and performed functional enrichment analysis. We postulate the potential functional roles of the two understudied kinases based on the enrichment analysis of their predicted substrates.

## MATERIALS & METHODS

KSFinder includes three major components, a knowledge graph built using data from different sources, knowledge graph embedding models that learn from the graph data, and an MLP classifier that discerns true kinase-substrate links (Fig. 1).

### Knowledge graph dataset

A knowledge graph dataset (Fig. 1A) constructed by *Gavali et al. (2022)* with 20 different relation types and 289,969 distinct entities, including (i) kinase-substrate phosphorylation interactions from iPTMnet (*Hongzhan et al., 2018*); (ii) molecular functions, biological processes, and their hierarchical relationships from Gene Ontology (GO) (*Ashburner et al., 2000*; *Gene Ontology Consortium, 2021*); (iii) associations of protein-pathway, protein-disease, protein-genetic disorder, disease-pathway, disease-genetic disorder, protein-complex, complex-pathway, and protein-complex relationships from BioKG (*Brian, Sameh & Vít, 2020*); and (iv) protein isoforms and ontology hierarchical relationships from Protein Ontology (PRO) (*Natale et al., 2011*) was adopted for this study (*Gavali et al., 2022*). This dataset contains 7,432 kinase-substrate relationships and 1,047,686 other relationships.
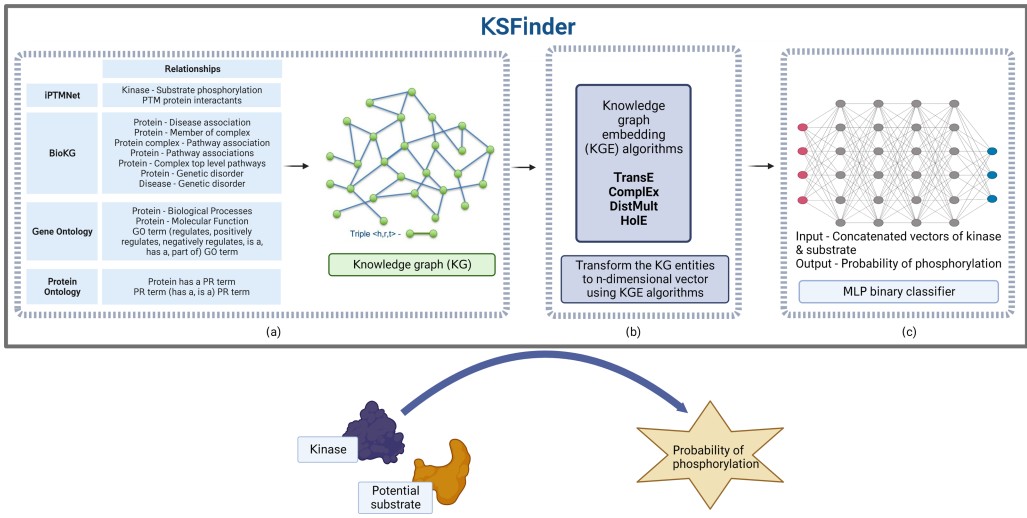

**Figure 1** **KSFinder - a knowledge-graph model for kinase-substrate link prediction.** (A) The different relationships of the kinases and substrates extracted from various resources are integrated to build a knowledge graph. (B) Knowledge graph embedding (KGE) algorithms are used to capture the semantics and assign vectors for the entities. (C) The vectors extracted from the KGE are used to build a multilayer perceptron (MLP) binary classifier that identifies kinase-substrate links. Created with BioRender.com.

## Knowledge graph embedding

Four different embedding models (Fig. 1B) were developed using the KGE algorithms TransE (*Antoine et al., 2013*), HolE (*Nickel, Rosasco & Poggio, 2016*), DistMult (*Bishan et al., 2014*), and ComplEx (*Théo et al., 2016*) leveraging the library provided by Ampligraph (*Luca et al., 2021*). The 7,432 kinase-substrate edges were split into training, validation, and test datasets in the ratio of approximately 60:20:20 respectively. The 4,632 kinase-substrate training triples were integrated with the other 1,047,686 triples. The validation set and test set had 1,400 triples each. Hyperparameter optimization was performed using the validation dataset to determine the optimal values for each of the model parameters—embedding size, eta (number of negatives), number of epochs, number of batches, learning rate, loss function, and regularizer. The values tested during parameter optimization are listed in Table 1. The negatives were generated using the closed-world assumption strategy (*Nickel et al., 2016*) by corrupting the head or tail entity during training. If the corruption resulted in a true triple, it was removed from the negatives. The optimal epoch for early stopping of the training was identified using the metric mean reciprocal rank (MRR). During validation, the positive triple was ranked along with the synthetic negatives by their algorithm scores. MRR was computed by averaging the reciprocal rank of the true triples in the validation dataset. The training was stopped when the MRR score did not increase for 3 consecutive evaluations. As we are interested in predicting potential substrates for the known human kinases, synthetic negatives for validation were generated by corrupting the tail entities (substrate proteins).

**Table 1  Hyperparameter optimization values for determining the best performing knowledge graph embedding model.**

| Hyperparameter | Values |
| --- | --- |
| Batch count | 8, 12, 16 |
| Embedding size | 90, 120, 150 |
| Number of negatives (eta) per positive triple | 10, 15, 20 |
| Loss function | pairwise, multiclass_nll |
| Regularization type | L1, L2, nuclear 3-norm |
| Learning rate | 0.0001, 0.001 |

## Kinase-substrate relation classifier

A multilayer perceptron (MLP)-based neural network classifier (Fig. 1C) was trained using the embedding vectors of the kinases and substrates from the positive and the negative sets with class labels of 1 and 0 respectively. For each triple, the embedding vector of the kinase and substrate was retrieved from the KGE model and concatenated. The data was tagged with an appropriate class label of 1 or 0. Of the four graph embedding algorithms, both ComplEx and TransE performed equally well. We chose embeddings from the ComplEx model as it can model one-to-many and both symmetric and asymmetric relationships. TransE, which captures only asymmetric relationships, performed well probably due to the fact that >99% of the relations in our network are asymmetric. The 4,632 kinase-substrate pairs from the training dataset along with 1,400 kinase-substrate pairs from the validation dataset used in the graph embedding model were used as the positive data for training the classifier.

Hyperparameter optimization was performed with 10-fold cross-validation to determine the optimal number of nodes in the hidden layer, the optimizer type, the alpha value for regularization, the learning rate, the activation function, and the maximum number of iterations. The model used binary cross-entropy loss function to minimize the loss and the rectified linear unit activation function, ReLU. The output from the neurons was transformed to probability using the softmax function.

$$\text{Binary Cross Entropy loss} = -\left(\frac{1}{n}\right)\sum_{i=1}^{n}\left(y_i * \log(p_i) + (1 - y_i) * \log(1 - p_i)\right)$$

where $y_i$ is the target class label; $p$ is the softmax probability; $p_i$ is the probability of the positive class, 1; and $(1 - p_i)$ is the probability of the negative class, 0.

$$\text{Softmax function} = \frac{e^{zi}}{\sum_{j=1}^{k} e^{zj}}$$

where $zi$ is the output from the neural network for class, $i$; $k$ is 2, the number of classes; $zj$ is the output for each class.

We retained the positive test set used in the graph embedding model as the test set for the classifier model. This was done to avoid potential data leaks from the embedding training to the classifier test set.

As there are no standardized datasets for negative kinase-substrate pairs, and because many true kinase-substrate relationships are unknown, compiling a high-quality negative dataset was challenging.

## PredKinKG negative dataset

To construct a negative dataset, PredKinKG represented the proteins in the embedding space by their cellular component annotations and computed the cosine similarity between them. Kinase-substrate pairs that were far away in the embedding space were selected as negatives because they are likely to be in different subcellular locations and therefore have a lower chance of interacting (*Gavali et al., 2022*). PredKinKG also included data from Negatome (*Blohm et al., 2014*), a database containing non-interacting protein pairs.

Though PredKinKG applied an effective strategy for generating the negatives, the triples generated by this approach were not a comprehensive dataset encompassing all the substrates in the positive group. Moreover, certain proteins were over-represented in the negative group. Because the classifier model is supervised and trained on the vectors of the substrates and kinases, an imbalanced representation of the entities in the two sets would inject unintended biases into the model's features and thereby the prediction results. Furthermore, the lack of true negatives makes it imperative to solve any prejudice in data for building an efficient prediction model.

## KS-negative dataset

We constructed an alternative negative dataset (KS-negative) that addresses the limitations of the PredKinKG negative dataset. The negative kinase-substrate pairs were generated by a combination of random sampling of kinase-substrate pairs along with selecting negatives from the PredKinKG dataset and Negatome.

The KS-negative dataset was compiled using the following process (Fig. 2):

1. As the Negatome protein pairs are experimentally supported, they were retained without any filtering. We observed that there were 16 pairs that overlapped between our positive set and Negatome. Each of these pairs and the literature citing them were reviewed manually and the false negative or false positive pairs were removed.

2. A hashmap was generated capturing the substrates and the counts of their representation among the positive triples. For every triple in the positive dataset, a random triple containing the same kinase was chosen from the PredKinKG's subcellular location based negative set. The count of triples with the same substrate as the selected triple in the negative dataset was compared with the count of the substrate in the hashmap. If the count of the chosen substrate in the negative exceeded that of the positive, the triple wasn't selected. This procedure ensured that the positive and negative sets had fairly equal counts of the same substrate protein. When this negative set was exhausted of triples, additional negative pairs were generated by random sampling detailed in the next step.

3. The list of proteins from the knowledge graph dataset was extracted and the phosphorylation scores for every kinase-substrate combination were computed using the ComplEx KGE model. When the comparative score of the triple is higher, it has

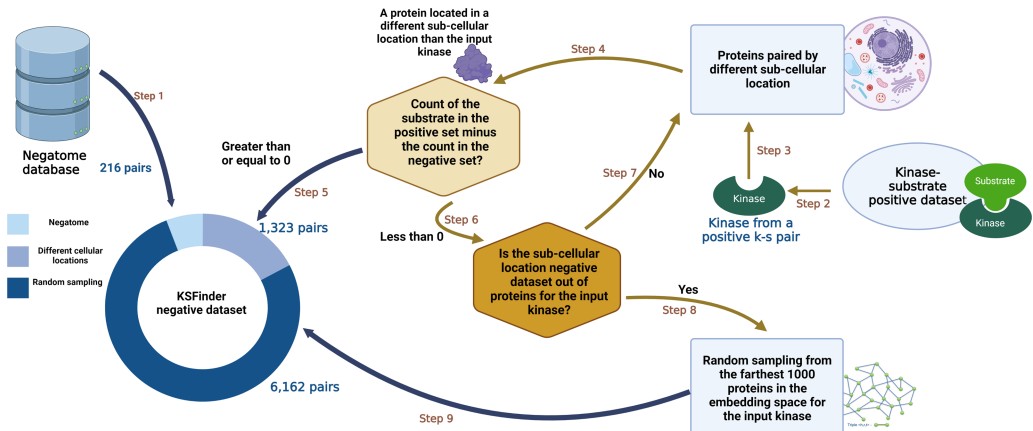

**Figure 2** **KSFinder negative data generation process.** (Step 1) Non-interacting protein pairs from Negatome dataset are added to the KSFinder negative dataset. (Step 2) From each kinase-substrate positive pair, the kinase is retrieved. (Step 3) The kinase is passed to the negatives generated by the subcellular location strategy (PredKinKG negatives). (Step 4) A negative pair containing the kinase is identified and the substrate protein is retrieved. (Step 5) If the count of the substrate protein in the positive dataset $\geq 0$, the negative pair is added to the KSFinder negative dataset. (Step 6) If the count of the substrate protein in the positive dataset $< 0$, the subcellular location strategy negatives are queried for additional negative substrates. (Step 7) If step 6 finds additional negative substrates for the kinase, steps 4 through 6 are repeated. (Step 8) If step 6 finds no additional negative substrates for the kinase, a protein is randomly sampled from the 1000 farthest proteins to the input kinase, from the KG embedding space. (Step 9) The negative pair is added to the KSFinder negative dataset. Created with BioRender.com.

a better chance of being a true phosphorylation triple. The substrate proteins were sorted in ascending order by their scores, and a random protein was selected from the first 1,000 proteins on the list. The selected substrate was tested for over-representation using the process detailed in Step 2.

The final negative dataset thus generated comprised 7,701 records, with pairs from Negatome, 1,323 pairs created *via* cellular location strategy, and 6,162 generated *via* random sampling. This dataset was split into training and test sets with counts of 6,301 and 1,400 respectively.

We will refer to this dataset as the 'KS-negative dataset' throughout the rest of the article.

## Evaluation of knowledge graph embedding models

We evaluated the embedding model independent of the downstream classifier. Two assessments were conducted to evaluate the performance of the four KGE models on two different datasets, PredKinKG and PredKinKG-B (balanced).

## Assessment 1—PredKinKG dataset

The PredKinKG's negative dataset was randomly sampled for two sets of data with 1,400 triples in each. One set was paired with the 1,400 positive validation triples and the other was paired with the 1,400 positive test triples. The embedding model assigns higher scores for triples that it predicts as positive and lower scores for negatives. Since the embedding model returns raw scores, they were calibrated using the Platt scaling method for conversion

to probability values. The best-performing KGE models from each of the four algorithms—TransE, ComplEx, DistMult, and HolE, as determined by the hyperparameter optimization were selected and assessed on the 2,800 test triples.

### Assessment 2—PredKinKG-B (balanced) dataset

As the negative set generated by PredKinKG over-represents certain entities, we generated a new dataset by sampling negative triples such that the representation of the entities (proteins) in the positive and negative sets was balanced. We refer to this dataset as the PredKinKG-B (balanced) dataset. This test set resulted in 662 triples with 331 in the positive set and 331 in the negative set. We evaluated the four graph embedding models on this subset.

### Evaluation of the integrated model (KSFinder)

KSFinder was evaluated on four different datasets (Assessment 3 through Assessment 6) to assess the model's generalization capability and ensure the model is not prejudiced towards certain datasets.

### Assessment 3—KSFinder dataset

KSFinder was evaluated on a test set containing 1,018 records with an equal proportion of positives and negatives. From the KS-negative test set, the pairs generated by random sampling were removed because the embeddings carry the semantics of the graph, and evaluating those pairs that were generated on the basis of embedding space may result in inflated scores. The resulting negative test set contained 96 pairs from Negatome and 413 pairs created *via* cellular location strategy. The positive dataset set used in Assessment 1 was under-sampled for 509 records.

### Assessment 4—PredKinKG-B dataset

We evaluated KSFinder on the PredKinKG-B (balanced) dataset used in Assessment 2 and compared its performance with the stand-alone graph embedding algorithm.

### Assessment 5—LinkPhinder dataset

The LinkPhinder benchmark dataset was downloaded from *Nováček et al. (2020)* to assess KSFinder's performance. Since LinkPhinder data is at the phosphorylation site level, a kinase-substrate relationship may exist in their positive and negative set with varying site information. As we are interested in the information at the kinase-substrate level, pairs that have at least one known phosphorylation site were selected for the positive set and those that have none were chosen for the negative set. The records that overlapped with KSFinder's training dataset were filtered out and KSFinder was evaluated on the remaining dataset. This test set contained 971 pairs each in the positive and negative sets.

### Assessment 6—PredKinKG dataset

KSFinder was evaluated on the PredKinKG dataset, the positive and negative test set used in Assessment 1.

### Comparison of KSFinder with other kinase-substrate prediction models

We evaluated the LinkPhinder (*Nováček et al., 2020*) and PredKinKG (*Gavali et al., 2022*) models, which report improved performance over other phosphorylation prediction tools, including NetPhospK, Scansite, NetworKIN, GPS, Phosphopredict, NetPhorest, and compared their performance with KSFinder.

### Assessment 7—Comparison with LinkPhinder

As LinkPhinder predicts site-specific phosphorylation activity, for a fair comparison with KSFinder, we extracted their prediction scores at the site level and computed the probability of the kinase phosphorylating the substrate at any site. The prediction score of a k-s pair was computed by,

$$\text{Phosphorylation probability of k} - s = \prod_{k=1}^{n}(1 - P(\text{Phos at k}))$$

where $n$ is the number of sites of phosphorylation, $k$ is the site number and $P(\text{Phos at }k)$ is the prediction probability reported by LinkPhinder at $k$.

Of these predictions, 2,021 kinase-substrate records overlapped with KSFinder's dataset. 1,027 were positive records and 994 were negative. Using the predicted probabilities reported by LinkPhinder and true labels from the KSFinder dataset, LinkPhinder's performance was assessed.

### Assessment 8—Comparison with PredKinKG

We did a direct comparison of PredKinKG with KSFinder using the prediction probability scores provided by PredKinKG. 14,370 predictions reported by PredKinKG overlapped with KSFinder's dataset. Of these, 7,017 were positive and 7,353 were negative.

For Assessment 7 and Assessment 8, we did not exclude the data that was used in the training set of LinkPhinder or PredKinKG, which gave an advantage to the other models over KSFinder, as they have a higher chance of accurately predicting the outcomes for the overlapping records. As no classification threshold was mentioned in the papers, we used the reported prediction probability values to compute the ROC-AUC and PR-AUC scores and compared them with the AUC values of KSFinder.

### Assessment 9—Comparison with Ser-Thr-KS atlas

*Johnson et al. (2023)* ranked the 303 serine/threonine kinases for the 89,784 human phosphosites. We selected the top 15 kinases for each substrate site and computed a unique set of positive predictions by pairing the kinases and substrates. The negative predictions were computed by pairing the proteins that ranked greater than 150. Since the reported ranks are at the site level, if a kinase scored a rank of 15 or less for at least one of the sites of the substrate, the kinase-substrate pair was retained only in the positive set and removed from the negative set. The pairs that overlapped with KSFinder's dataset were retained. This resulting dataset contained 3,848 positives and 3,708 negatives.

## Literature mining for evidence attribution

We employed KSFinder to predict substrates of the 432 kinases in our knowledge graph, 68 of which are listed as dark kinases by IDG. To further assess the prediction outcomes from KSFinder, we mined the literature for evidence attribution of predictions for 68 understudied kinases. Because curation of the literature is not comprehensive, we anticipated that some kinase-substrate relationships that are not present in our dataset but are predicted by our model may be reported in the literature. We considered kinase-substrate predictions with a probability score >= 0.7 to be positive predictions and those with scores <= 0.3 to be negative predictions. RLIMS-P, a rule-based phosphorylation information extraction text mining tool (*Torii et al., 2015*), was used to extract kinase and substrate mentions and relationships from literature in PubMed Central and Medline.

The process for automated query construction and information retrieval and for manual evidence evaluation is summarized below:

1. The gene names including the synonyms of the proteins in our dataset were retrieved from UniProt.
2. For every prediction pair, a query was constructed.
3. The query contains two parts, the kinase gene name and substrate gene name separated by "AND", in the format "(KINASE GENE) AND (SUBSTRATE GENE)"
4. If the protein had multiple gene synonyms, each name was separated by an "OR", in the format "(KINASE GENE NM1 OR KINASE GENE NM2) AND (SUBSTRATE GENE NM1 or SUBSTRATE GENE NM2)".
5. The constructed query was passed as input to the RLIMS-P tool using the iTextMine REST API, https://research.bioinformatics.udel.edu/itextmine/api/search/query/rlims (*Ren et al., 2018*).
6. Both PMC and Medline were queried with the input query.
7. The response returned by the API was in JSON format. This JSON response was parsed to retrieve kinase names and substrate names. If the response had at least one matching kinase gene name and substrate gene name, the prediction was considered valid and selected for manual review.
8. The positive results from literature mining were reviewed manually to evaluate the results and provide evidence attribution. We manually identified evidence for a few dark kinases.

## Cases study: enrichment analysis for the understudied kinases—HIPK3 and CAMKK1

The substrates predicted to be phosphorylated by the two kinases, HIPK3 (homeodomain interacting protein kinase 3) and CAMKK1 (calcium/calmodulin-dependent protein kinase kinase 1) with high confidence were retrieved by setting the probability threshold cut-off value of 0.9. Enrichment analysis was performed on the selected substrates using the tool, DAVID (Database for Annotation, Visualization, and Integrated Discovery tool) (*Sherman et al., 2022*). As our model had identified the potential substrates from a fraction of the known human proteins that are in our knowledge graph, the proteins in the knowledge graph were used as the background instead of all human proteins for the enrichment analysis

study. The top enriched GO terms in the three categories, biological process, molecular function, and cellular component, were selected by Benjamini–Hochberg corrected *p*-value threshold of 0.1 and by highest fold enrichment. We postulate the potential roles of the kinases by reviewing their enriched terms and presenting results in the literature supporting the proposed functions and processes.

### Comparison of results from KSFinder and Ser-Thr-KS atlas

We retrieved a unique set of substrates that scored a rank of 15 or lower for HIPK3 and CAMKK1 separately from the Ser-Thr-KS atlas and compared it with the high-confidence substrates identified by KSFinder.

## RESULTS

### Knowledge graph embedding
#### *KGE model hyperparameter optimization and performance*

The optimal hyperparameters determined by the grid search technique for the four best performing KGE models are provided in Table 2.

All the models performed the best at an embedding size of 150, at a learning rate of 0.0001, and with a loss function of multiclass negative loss likelihood. The performance metrics of the KGE models on the validation data set are shown in Table 3.

The optimal parameters of the best-performing model were determined using the MRR. TransE scored the highest MRR of 0.1006 and ComplEx ranked next with a score of 0.0923. Hits@n denotes the proportion of positive triples in the validation dataset (total of 1,400) that scored at least nth rank relative to the negative triples. The TransE algorithm uses the sum of head and relation vectors to approximate the vector of the tail entity. Though TransE captures only asymmetric relationships, it exceeded the MRR score of ComplEx which captures both symmetric and asymmetric relationships by approximately 0.01. This could be due to the fact that less than 1% of the relations in our knowledge graph are symmetric. The DistMult algorithm is similar to ComplEx in its scoring function, where it uses the dot product of the head and relation to approximate the tail vector. However, DistMult captures only symmetric relationships and therefore scores relatively lower on our KG. Holographic Embedding (HolE) is known to capture compositional relations, yet its best-performing model scored lower than the other three embedding algorithms and was not optimal for our dataset. These metrics compare the performance of the four KGE algorithms on the validation dataset using the synthetic negatives generated by the CWA corruption strategy.

#### *Evaluation of KGE*

Assessment 1 and Assessment 2 evaluate the performance of the four KGE models on the PredKinKG and the PredKinKG-B datasets, respectively. In agreement with the MRR scores on the validation dataset, the performance of both TransE and ComplEx is superior to the DistMult and HolE models on the PredKinKG dataset (Table 4). Additionally, the scores of the ComplEx model were close to those of the TransE model. Similarly, the ComplEx model has the highest ROC-AUC of 0.751 and is the second in PR-AUC with a score of

**Table 2  Optimal values for the KGE hyperparameters determined by the validation dataset.**

| Hyperparameter | ComplEx | TransE | DistMult | HolE |
|---|---|---|---|---|
| Batch count | 12 | 12 | 12 | 16 |
| Embedding size | 150 | 150 | 150 | 150 |
| Number of negatives (eta) per positive triple | 15 | 10 | 10 | 20 |
| Loss function | multiclass_nll | multiclass_nll | multiclass_nll | multiclass_nll |
| Regularization type lambda=1e−05 | L1 | L1 | L1 | nuclear 3-norm |
| Optimizer (Adam) Learning rate | 0.0001 | 0.0001 | 0.0001 | 0.0001 |
| Epochs | 4500 | 2700 | 7200 | 5100 |

**Table 3  Performance of the KGE models on the validation dataset.**

| KGE model | Hits@1 | Hits@3 | Hits@10 | Mean Reciprocal Rank (MRR) |
|---|---|---|---|---|
| TransE | 0.0714 | 0.0943 | 0.1564 | 0.1006 |
| ComplEx | 0.0614 | 0.0886 | 0.1386 | 0.0923 |
| DistMult | 0.0479 | 0.0814 | 0.1288 | 0.0801 |
| HolE | 0.0379 | 0.0679 | 0.1336 | 0.0731 |

**Table 4  ROC-AUC and PR-AUC of the four knowledge graph embedding models on PredKinKG and PredKinKG-B datasets.**

| Assessment | Dataset | Metrics | TransE | DistMult | ComplEx | HolE |
|---|---|---|---|---|---|---|
| Assessment 1 | PredKinKG | ROC-AUC | 0.868 | 0.861 | 0.872 | 0.856 |
| | | PR-AUC | 0.871 | 0.848 | 0.869 | 0.85 |
| Assessment 2 | PredKinKG-B | ROC-AUC | 0.74 | 0.742 | 0.751 | 0.739 |
| | | PR-AUC | 0.713 | 0.698 | 0.718 | 0.723 |

0.718 when evaluated using the PredKinKG-B dataset (Table 4). It should be noted that the performance of all four graph embedding models dropped significantly when evaluated on the PredKinKG-B dataset. The models were good at predicting the relations involving the pool of proteins in the PredKinKG's biased negative set but did not generalize well and scored lower when evaluated on the more balanced dataset.

The ComplEx KGE model performed better than TransE on the PredKinKG-B dataset, and nearly equal on the PredKinKG dataset. Given the fact that ComplEx models one-many relationships and learns semantics irrespective of the relation symmetry, and its relatively superior performance over other KGE models, the vectors embedded by the ComplEx KGE were selected as features for developing the downstream classification model.

## Multilayer perceptron classification
### Hyperparameter optimization
The training data for the classifier model consisted of 12,333 records with 6,032 positives and 6,301 negatives. The 6,043 positives were obtained by combining the training data

**Table 5  Optimal values for the MLP classifier determined by cross-validation.**

| Hyperparameter | Values |
|---|---|
| Alpha (L2 regularization term value) | 0.0001 |
| 1 hidden layer with node count | (40,) |
| Activation function | ReLU |
| Solver | Adam |
| Learning rate | Constant (0.0001) |

(4,632) and validation data (1,400) used in the graph embedding model. The number of nodes and optimal hyperparameters at which the MLP classifier performed the best are summarized in Table 5. These parameter values were determined *via* the grid search technique and 10-fold cross-validation of the training data. KSFinder used ReLU, the rectified linear unit activation function. The prediction probability for phosphorylation was computed by transforming the output from the neural network for label 1 using the softmax function.

### Evaluation of KSFinder

Figure 3 shows the performance plots of KSFinder on the four different datasets as described in Assessments 3, 4, 5, and 6. KSFinder exhibits consistent performance on KSFinder dataset, PredKinKG, and PredKinKG-B datasets with a ROC-AUC value ranging from 0.8 to 0.83 (Fig. 3A), and PR-AUC ranging from 0.795 to 0.85 (Fig. 3B). It demonstrates an acceptable discriminative capability on the LinkPhinder dataset with a ROC-AUC of 0.747 (Fig. 3A) and PR-AUC of 0.722 (Fig. 3B).

In comparison with the performance of the stand-alone ComplEx KGE model (Assessment 2) on the PredKinKG-B dataset, KSFinder showed improved performance on the same dataset (Assessment 4). These results also demonstrate KSFinder's relatively better generalization capability on different datasets than the KGE model.

### Comparative assessment results of KSFinder with LinkPhinder, PredKinKG, and Ser-Thr-KS atlas

Figure 4 shows the performance of KSFinder, LinkPhinder, PredKinKG, and Ser-Thr-KS. KSFinder outperforms all three prediction models in ROC-AUC (Fig. 4A) and PR-AUC (Fig. 4B). A challenge with comparative assessment is the models are trained on different datasets, so a direct comparison was not possible. However, we tried to perform an impartial comparison by not filtering their training data in the prediction scores reported by LinkPhinder and PredKinKG. Additionally, we assessed KSFinder on LinkPhinder and PredKinKG datasets. This two-way comparison provided the opportunity for better judgment of KSFinder's generalized performance, which is a shortfall in other prediction models. As the Ser-Thr-KS atlas is not built on true labels, we could not compare KSFinder's performance on their dataset.
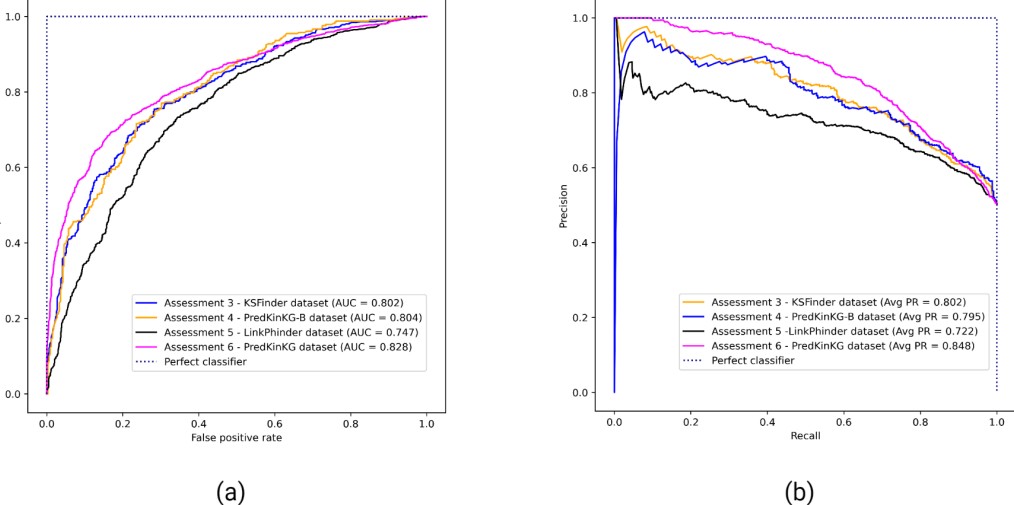

**Figure 3  Performance of KSFinder on different datasets.** Each curve indicates the performance of KS-Finder in terms of (A) ROC-AUC and (B) PR-AUC on the four datasets at different thresholds from 0 through 1. The label denotes the assessment number, the dataset, and the corresponding curve.

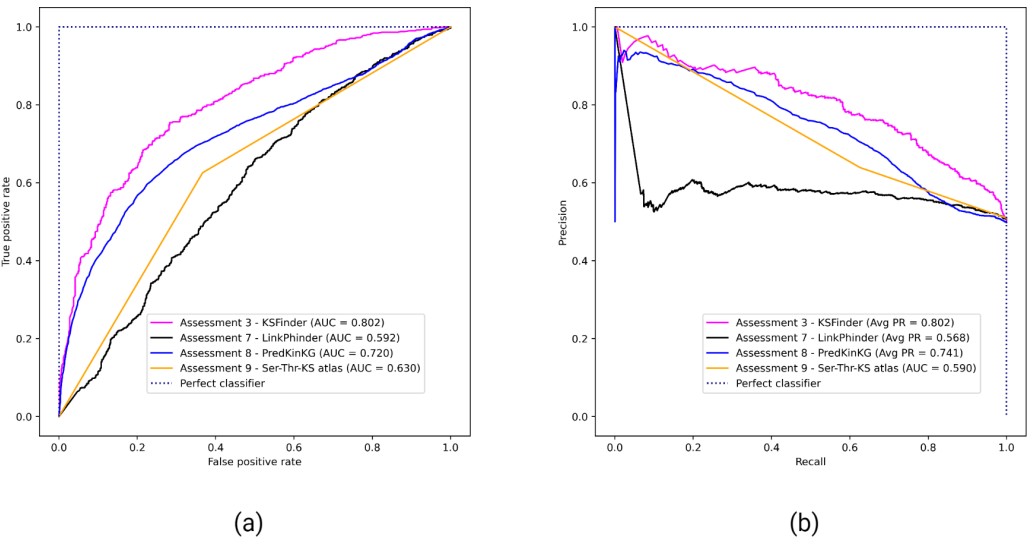

**Figure 4  Performance of KSFinder, LinkPhinder, PredKinKG, and Ser-Thr-KS atlas on KSFinder dataset.** Each curve indicates the performance of the four models in terms of (A) ROC-AUC and (B) PR-AUC on the KSFinder dataset at different thresholds from 0 through 1. The label denotes the assessment number, the model, and the corresponding curve.

## Functional analysis
### *Literature evidence of the predictions*
We found literature evidence supporting 17 of our 36,055 positive predictions (score >0.7) involving the 68 understudied kinases. Of these, nine scored a probability value greater than 0.9, six a probability value between 0.8 and 0.9, and two a probability value between

0.7 and 0.8. Three of the 17 were identified *via* a manual search of literature when studying HIPK3 and NUAK2 kinases, whereas the remaining 14 were identified by RLIMS-P tool and manually reviewed for phosphorylation activity. The evidence of the predictions is summarized in Table 6.

### False negative predictions

93,593 of KSFinder's predictions with a probability value <= 0.3 were assessed and RLIMS-P identified evidence for four of them. These include the autophosphorylation of HIPK1 (*Ecsedy, Michaelson & Leder, 2003*), phosphorylation of MEK1 by BCKDK (*Xue et al., 2017*), BCAR1 by PKN3 (*Gemperle et al., 2019*) and HEY1 by STK38L (*López-Mateo et al., 2016*). These positive phosphorylation pairs were incorrectly classified as negatives by KSFinder.

Of the total 21 phosphorylation pairs identified in the literature, our model identified 17 true positives and 4 false negatives. Excluding the 3 that were identified *via* manual search from the 17 true positives and the total, KSFinder scores a sensitivity of 0.78.

## Case study

### Enrichment analysis of HIPK3

Of the proteins predicted to be phosphorylated by HIPK3, 230 scored a probability value greater than 0.9. Functional enrichment analysis of the 230 proteins suggests HIPK3'ss potential role in the extracellular matrix organization and epigenetic regulation of gene expression (Table 7).

The top enriched biological processes are the intracellular steroid hormone receptor signaling pathway, positive regulation of extracellular matrix organization, epigenetic regulation of gene expression, and cellular responses to chemical stimuli. The top enriched molecular functions of HIPK3 include BH3 domain binding, and cysteine-type endopeptidase activity involved in the apoptotic signaling pathway which supports HIPK3's role in apoptosis. NAD-dependent histone deacetylase activity and primary miRNA binding align with the protein's proposed role in epigenetic gene expression. The enriched cellular components support the protein's role in transcription regulation and cellular response to chemical stimuli.

## Enrichment analysis of CAMKK1

CAMKK1 is a protein belonging to the calcium-triggered signaling cascade involved in a number of cellular processes. A total of 126 substrates were predicted to be phosphorylated by CAMKK1 with a probability value greater than 0.9. The functional enrichment results of CAMKK1 substrates are summarized in Table 8.

The top enriched biological process of CAMKK1 includes synapse organization and negative regulation of neuron death, glucose homeostasis, regulation of lipid storage, and fatty acid oxidation. These suggest CAMKK1's potential role in the regulation of cellular metabolism and involvement in neuronal development. Tetrahydrobiopterin binding is the top enriched molecular function of CAMKK1. Tetrahydrobiopterin is an enzyme cofactor that carries electrons in redox reactions. This molecular function correlates with

**Table 6  Evidence of KSFinder's novel predictions in the scientific literature.**

| Kinase [UniProt ID] | Substrate [UniProt ID] | Prediction probability | Information in Literature | Method |
|---|---|---|---|---|
| HIPK3 [Q9H422] | RUNX2 [Q13950] | 0.98787 | Experimental results support HIPK3 phosphorylation of Runx2 AD3 residues (313-332) in vitro. Phosphorylation of Jun and Runx2 by HIPK3 negatively regulates apoptosis and enhances androgen receptor-mediated transcriptional activation (*Guo et al., 2022*; *Sierra & Towler, 2010*) | Manual Search |
| HIPK3 [Q9H422] | STAT3 [P40763] | 0.98445 | Phosphorylation of STAT3 plays a critical role in the inflammatory response of Allergic Conjunctivitis, and STAT3 phosphorylation could be downregulated by inhibiting HIPK3 expression (*Guo et al., 2022*) | Manual Search |
| DYRK3 [O43781] | SIRT1 [Q96EB6] | 0.9806 | DYRK3 promotes cell survival through direct phosphorylation of SIRT1 at Thr(522) residue and promotes deacetylation of p53 (*Guo et al., 2010*) | RLIMS-P |
| NUAK2 [Q9H093] | MYPT1 [O14974] | 0.95275 | Myosin phosphatase target subunit 1 (MYPT1) was identified as a specific substrate for NUAK2 using an in vitro kinase assay and high-performance liquid chromatography (HPLC). The literature data suggests NUAK2 phosphorylation of MYPT1 at different sites elicits various regulatory functions (*Yamamoto et al., 2008*) | RLIMS-P |
| DYRK1B [Q9Y463] | NKX3.1 [Q99801] | 0.94169 | DYRK1B phosphorylates the tumor suppressor gene, NKX3.1 at serine 185 residue, and this is experimentally validated by in-vitro kinase assay (*Song et al., 2015*) | RLIMS-P |
| MARK4 [Q96L34] | MAP2 [P11137] | 0.92479 | MARK4 phosphorylation of MAP2 and MAP4 regulates cell polarity (*Ahrari, Mogharrab & Navapour, 2020*) | RLIMS-P |
| TLK1 [Q9UKI8] | TLK1 [Q9UKI8] | 0.91576 | Tousled-like kinases are an evolutionarily conserved family of proteins. The article demonstrates the autophosphorylation of TLK1 in Trypanosoma brucei, a unicellular protozoan parasite, and its interaction with AUK1, Asf1A, and Asf1B *in vivo* (*Li, Gourguechon & Wang, 2007*) | RLIMS-P |
| TTBK1 [Q5TCY1] | CRMP2 [Q16555] | 0.90949 | Experimental results show that TTBK1 in HEK293 cells induced significant Collapsin response mediator protein-2 (CRMP2) phosphorylation at T514. This phosphorylation may cause neurite degeneration and somal accumulation of pTau protein in Alzheimer's disease (*Ikezu et al., 2020*) | RLIMS-P |
| TLK2 [Q86UE8] | TLK2 [Q86UE8] | 0.90513 | The coiled-coil domains of TLK2 mediate dimerization and are essential for the auto-phosphorylation of TLK2 (*Mortuza et al., 2018*) | RLIMS-P |
| PRKACG [P22612] | GJA1, Connexin-43 (Cx43) [P17302] | 0.88295 | "PKCgamma phosphorylates Cx43 on serine and this causes disassembly and loss of gap junction Cx43 from the cell surface." (*Wagner et al., 2002*) | RLIMS-P |
| PAK5 [Q9P286] | SATB1 [Q01826] | 0.85906 | PAK5 phosphorylates SATB1 on Serine 47 residue and this initiates the epithelial-mesenchymal transition cascade and regulates metastasis of cervical cancer cells (*Huo et al., 2019*) | RLIMS-P |
| MARK4 [Q96L34] | MAP4 [P27816] | 0.85578 | MARK4 phosphorylation of MAP2 and MAP4 regulates cell polarity (*Ahrari, Mogharrab & Navapour, 2020*) | RLIMS-P |

**Table 6** (*continued*)

| Kinase [UniProt ID] | Substrate [UniProt ID] | Prediction probability | Information in Literature | Method |
|---|---|---|---|---|
| TAOK2 [Q9UL54] | MKK4 [P45985] | 0.85555 | "TAOK2 was found to phosphorylate MKK4/MKK7 and activate the JNK signaling cascade." (*Fang et al., 2020*) | RLIMS-P |
| CDK10 [Q15131] | PKN2 [Q16513] | 0.84719 | CDK10 phosphorylates PKN2 on threonine 121 and 124 in PKN2's core RhoA-binding domain, which is essential for the stabilization of RhoA protein and the actin network architecture (*Guen et al., 2016*) | RLIMS-P |
| MLK2 [Q02779] | SEK1 [P45985] | 0.83275 | "Recombinant MST/MLK2 produced in bacteria directly phosphorylates and activates SEK1/MKK4/JNKK in vitro" (*Hirai et al., 1997*) | RLIMS-P |
| NUAK2 [Q9H093] | MLC2 [P10916] | 0.78413 | Phosphorylation of MLC2 by NUAK2 is required for the proper functioning of myosin in the developing neural tube; loss of NUAK2 activity has been associated with anencephaly, a birth defect in which the brain and skull do not develop fully (*Bonnard et al., 2020*) | Manual Search |
| STK3 [Q9Y6E0] | LATS1 [O95835] | 0.75175 | STK3 phosphorylation of LATS1 is part of the cascade events in the Hippo signaling pathway (*Moon et al., 2019*) | RLIMS-P |

**Table 7** Enrichment analysis of top predicted phosphorylation substrates of HIPK3 kinase.

| Category | GO ID | Term description | Fold enrichment | False discovery rate |
|---|---|---|---|---|
| Biological Process | GO:0030518 | Intracellular steroid hormone receptor signaling pathway | 11.946 | 1.73E−02 |
| | GO:1903055 | Positive regulation of extracellular matrix organization | 11.946 | 8.43E−02 |
| | GO:1901522 | Positive regulation of transcription from RNA polymerase II promoter involved in the cellular response to chemical stimulus | 9.955 | 3.58E−02 |
| | GO:0040029 | Regulation of gene expression, epigenetic | 8.959 | 8.80E−07 |
| | GO:0035067 | Negative regulation of histone acetylation | 8.533 | 6.30E−02 |
| | GO:0001836 | Release of cytochrome c from mitochondria | 7.964 | 1.72E−03 |
| Molecular Function | GO:0051434 | BH3 domain binding | 12.008 | 2.03E−02 |
| | GO:0005496 | Steroid binding | 9.607 | 4.25E−02 |
| | GO:0097199 | Cysteine-type endopeptidase activity involved in apoptotic signaling pathway | 8.006 | 7.50E−02 |
| | GO:0070878 | Primary miRNA binding | 7.505 | 2.26E−02 |
| | GO:0032041 | NAD-dependent histone deacetylase activity (H3-K14 specific) | 7.505 | 2.26E−02 |
| Cellular Component | GO:0035976 | Transcription factor AP-1 complex | 12.046 | 4.46E−02 |
| | GO:0071141 | SMAD protein complex | 9.637 | 8.63E−02 |
| | GO:0071159 | NF-kappaB complex | 9.637 | 8.63E−02 |
| | GO:0017053 | Transcriptional repressor complex | 5.815 | 5.34E−06 |
| | GO:0005667 | Transcription factor complex | 5.163 | 3.69E−17 |

**Table 8  Enrichment analysis of top predicted phosphorylation substrates of CAMKK1 kinase.**

| Category | GO ID | Term description | Fold enrichment | False discovery rate |
|---|---|---|---|---|
| Biological Process | GO:0010883 | Regulation of lipid storage | 22.754 | 5.89E−02 |
| | GO:0019395 | Fatty acid oxidation | 22.754 | 5.89E−02 |
| | GO:0002028 | Regulation of sodium ion transport | 13.652 | 1.01E−02 |
| | GO:1901215 | Negative regulation of neuron death | 9.102 | 5.06E−03 |
| | GO:0050808 | Synapse organization | 9.102 | 5.89E−02 |
| | GO:0042593 | Glucose homeostasis | 7.001 | 8.55E−04 |
| Molecular Function | GO:0034617 | Tetrahydrobiopterin binding | 22.873 | 3.34E−02 |
| | GO:0015631 | Tubulin binding | 5.967 | 9.65E−02 |
| | GO:0005080 | Protein kinase C binding | 5.003 | 8.76E−02 |
| | GO:0061629 | RNA polymerase II sequence-specific DNA binding transcription factor binding | 3.202 | 3.34E−02 |
| | GO:0005516 | Calmodulin binding | 2.938 | 4.48E−02 |
| Cellular Component | GO:0016234 | Inclusion body | 9.521 | 4.13E−02 |
| | GO:0012506 | Vesicle membrane | 8.16 | 5.76E−02 |
| | GO:0031966 | Mitochondrial membrane | 6.664 | 4.13E−02 |
| | GO:0005901 | Caveola | 4.998 | 5.76E−02 |
| | GO:0005741 | Mitochondrial outer membrane | 4.197 | 4.13E−02 |
| | GO:0098794 | Post synapse | 4.032 | 4.13E−02 |

the proposed role in cellular metabolism. In concordance with the diverse potential roles postulated for the kinase, the subcellular location of CAMKK1 is also diverse.

## Comparison of results from KSFinder and Ser-Thr-KS

The high-confidence predictions from KSFinder for HIPK3 and CAMKK1 were compared with the results from the Ser-Thr-KS atlas. Despite the fact that Ser-Thr-KS atlas captures only direct relationships, and KSFinder uncovers direct and indirect links, approximately 50% of the substrates predicted by KSFinder overlapped with the substrates scored by the Ser-Thr-KS atlas for HIPK3 and CAMKK1. Per the Ser-Thr-KS atlas, 4,629 phosphosites are predicted to be potentially phosphorylated by HIPK3. This comprises 2,624 unique substrates and overlaps with 110 of the 230 high-confidence HIPK3 substrates predicted by KSFinder. Similarly, for CAMKK1, 64 of the 126 high-confidence substrates predicted by KSFinder overlaps with 3,650 substrates identified by Ser-Thr-KS atlas.

## DISCUSSION

In this work, we presented our predictive model built using the integration of knowledge graph embedding and a neural network for studying kinase-substrate relationships. We showed that our model has superior performance over the existing prediction models. Knowledge graph representation learning has the advantage of capturing implicit relations in a complex network which is often overlooked by local feature-based selection techniques. Since the kinase-substrate interaction happens in a complex network of cellular processes,

this learning technique has the capability of discovering new links that other approaches fail to uncover. While the embedding method itself achieves good predictive performance, it cannot generalize on a stringent dataset. We took advantage of the data captured in its embedding vectors to train a classification model using a neural network and showed that the overall prediction performance increased when the embedded vectors were utilized as input features for the final prediction model.

We demonstrated the robustness of our model's performance through validation with several different test sets. Though the positive sets used by the different tools overlap, each tool uses a unique strategy for negative data generation. The performance scores and predictions by the different models may be biased based on their negative generation strategy. This study showed that our model's performance was consistent across datasets, demonstrating its generalization capability.

KSFinder uses high-level information about the proteins and has the potential to identify direct and indirect phosphorylation links. The lack of an identifier in the underlying dataset and the ambiguity stemming from the literature sources makes the classification of direct and indirect phosphorylation, a challenging task. This ambiguity in identifying the primary kinase-substrate links in KSFinder's training dataset is in its prediction results too. Nonetheless, the indirect predictions made by KSFinder are valuable for our goal of hypothesizing the functions of understudied kinases. The other kinase-substrate prediction models, aside from PredKinKG do not use the high-level interaction information and may be suited less ideally for kinase functional enrichment analysis case study.

KSFinder provides substrate predictions for 432 unique kinases, a much wider range of kinases than other tools except PredKinKG and we demonstrate improved performance of KSFinder over PredKinKG.

To evaluate the validity of the predictions generated by our model, we mined the literature for evidence of positive and negative predictions and reported a sensitivity score of 0.78 with 14 true positives and 4 false negatives. The high recall of KSFinder in text-mining evaluation demonstrates its potential to identify positive phosphorylation links.

HIPK3 is a serine/threonine protein kinase involved in transcription regulation, apoptosis, and steroidogenic gene expression. Our analysis hints at the protein's role in extracellular matrix organization. *Deng et al. (2019)* demonstrate the upregulation of pre-HIPK3 in response to adrenaline *via* the transcription factor, CREB1. The enriched molecular function and cellular component terms align with the protein's role in apoptosis. Though the apoptotic process is directly linked to HIPK3 in our knowledge graph, there are no explicit links connecting ECM regulation or epigenetic gene expression regulation roles to HIPK3.

Functional enrichment analysis of CAMKK1 shows the protein's potential involvement in synapse organization and negative regulation of neuronal death. In concordance with our results, the over-expression of CAMKK1 is linked to neurite outgrowth and axon regeneration in the peripheral nerves (*Zhao et al., 2023*). Though the only processes directly linked to CAMKK1 in our KG are protein phosphorylation, intracellular signal transduction, and positive regulation of kinase activity, our analysis connects CAMKK1

with neuronal organization. Similarly, we found evidence supporting CAMKK1's role in glucose uptake in skeletal muscles, a function relating to the proposed role of CAMKK1 in glucose homeostasis (*Carol et al., 2007*).

A limitation of our model is that it has been trained only on approximately 13% of the known human proteins, which comprise the known kinases, their phosphorylation substrates, and their interacting protein partners. Therefore, our model can perform link predictions for kinase-substrate relations involving only these proteins in the knowledge graph. Additionally, though our model utilizes the functional properties of the kinases and substrates, it does not consider the sequence, structure, or site-specific information of the proteins. In a future study, embeddings generated by capturing sequence and structural data along with the functional information of the proteins may be used to predict site-specific phosphorylation residues and reveal novel links in the kinome-phosphorylation site graph.

## CONCLUSIONS

Here, we show that an integrated model built using knowledge graph embedding and a neural network classifier predicts kinase-substrate relations with high sensitivity. This approach first captures the semantics in the protein phosphorylation network in its embedding vectors and then uses the vectors to train a binary classifier and discern kinase-substrate pairs. The integrated model not only outperforms the stand-alone KGE model but also exhibits robust performance on different datasets. It also shows superior performance over other kinase-substrate prediction tools and provides prediction coverage for 432 human kinases of which 68 are understudied. Future work will aim at expanding the protein network which is currently limited to 13% of the human proteins and at including sequence and structural information in the embedded vectors.

### Funding

This work was funded by grant number R35GM141873 from the National Institute of General Medical Sciences and grant number U01CA239106 from the National Cancer Institute of the National Institutes of Health, and supported by the computing resources funded by grant number 1919839 from the National Science Foundation. The funders had no role in study design, data collection and analysis, decision to publish, or preparation of the manuscript.

### Grant Disclosures

The following grant information was disclosed by the authors:
National Institute of General Medical Sciences: R35GM141873.
National Cancer Institute of the National Institutes of Health: U01CA239106.
The National Science Foundation: 1919839.

### Competing Interests

The authors declare there are no competing interests.

## Author Contributions

- Manju Anandakrishnan conceived and designed the experiments, performed the experiments, analyzed the data, prepared figures and/or tables, authored or reviewed drafts of the article, and approved the final draft.
- Karen E. Ross conceived and designed the experiments, performed the experiments, analyzed the data, authored or reviewed drafts of the article, and approved the final draft.
- Chuming Chen conceived and designed the experiments, authored or reviewed drafts of the article, and approved the final draft.
- Vijay Shanker conceived and designed the experiments, authored or reviewed drafts of the article, and approved the final draft.
- Julie Cowart analyzed the data, authored or reviewed drafts of the article, and approved the final draft.
- Cathy H. Wu conceived and designed the experiments, authored or reviewed drafts of the article, and approved the final draft.

## Data Availability

The data, models & predictions are available at Zenodo: Anandakrishnan Manju. (2023). KSFinder_KG_data. https://doi.org/10.5281/zenodo.7856947.

The code is available at GitHub and Zenodo:

– https://github.com/manju-anandakrishnan/ksfinder/releases/tag/v1.5.

– Manju A. (2023). manju-anandakrishnan/ksfinder: v1.5 (v1.5). Zenodo. https://doi.org/10.5281/zenodo.8134698.

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
