# Peer review of "KSFinder—a knowledge graph model for link prediction of novel phosphorylated substrates of kinases"

_PeerJ, doi:10.7717/peerj.16164_

## Round 0.1 · original submission · Major Revisions

Thanks again for your patience. The reviewers are generally favorable of the work, though bring up a number of questions/issues. In particular, questions around access to code, determination of the false positive rate, dataset bias, and some methodological questions and relationship to other methods are brought up. Please review the detailed feedback provided by the reviewers and address their concerns in your revision.

·

Basic reporting

(My sincere apologies to the authors for the delay in completing this review—hopefully these comments will prove useful.)

In this paper the authors describe a computational method for predicting novel kinase-substrate interactions that involves first generating embeddings from a knowledge graph with functional information about kinases and substrates, then training a supervised model to predict phosphorylation interactions from the knowledge graph embedding features. The authors highlight three features of the approach that are novel compared to prior knowledge-graph based approaches (including their own method, PredKinKG): 1) use of a neural network rather than a random forest classifier for the supervised learning step; 2) an approach to generating negative examples for supervised model training that reduces dataset imbalances; and 3) refinements to the algorithms used for generating the knowledge graph embeddings. The performance of the method is validated by comparing predicting performance against two other KG-based methods for phosphorylation prediction and by evaluating novel predictions for two understudied kinases.

Overall the study is a valuable contribution to the study of phosphorylation networks, however a few revisions would make it stronger.

Major points:

1. The paper should include links to a) code used to build the knowledge graph and train the supervised models, and b) the full set of predictions generated by the supervised KSFinder model. Predictions made from other models used to calculate the scores for the various assessment would also be valuable for anyone seeking to reproduce the assessments, but a) and b) are essential.

2. The method of generating negatives used in the paper uses (in part) low knowledge graph link prediction scores from the ComplEx KGE model (lines 252-258). Depending on how many negatives are generated this way, this could lead to bias in evaluation of the full KSFinder model. This is because the supervised model is using the ComplEx model embeddings as input, and these embeddings, by construction, can distinguish these low-scoring negative examples. It is not clear from the way the method is described what data the ComplEx KGE model that was used to generate the negatives was trained on. Even using the KGE model trained only on the training data to generate the negatives could be problematic as it would result in negatives associated with the training set being sorted into validation and test. The fact that the full KSFinder model performs so much better on its own dataset than any other dataset, including PredKinKG, which explicitly did not use this new negative generation method, makes such bias seem probable.

Given that the high performance of KSFinder and the novel approach to generating negative examples are key claims of the paper, any potential dataset bias should be clearly addressed. The authors could 1) perform their assessments using only steps 1-2 of negative construction to see if the KGE-generated negatives impact their results, or 2) structure the generation of negatives from the KGE model to guarantee no information leakage into validation and test; and 3) be explicit about the composition of different types of negative examples in the dataset. As a diagnostic, it may be informative to use a different embedding model to generate negatives (e.g., TransE) than the one used as input to the supervised model, though this may not be sufficient to eliminate bias (depending on degree of similarity in the learned representations).

3. For the comparison with LinkPhinder, a site-specific prediction method, the authors note on lines 313-319 that “we…chose the highest probability value for every kinase-substrate pair.” Instead of using the maximum probability, it would be more justifiable to use the combined probability of *any* of the site-specific phosphorylations being true (phos at site 1 OR phos at site 2 OR phos at site 3, etc). A protein with many sites, each with a moderate probability of any individual site being phosphorylated, might therefore have a high cumulative probability of being phosphorylated at the protein level. This could be calculated from the LinkPhinder scores as

1 - Product_{over all i} (1 - probability_of_phosphorylation(site_i))

In other words, 1 minus the probability of no phosphorylation on any site on the protein. The LinkPhinder results for Assessment 7 should be calculated using this approach and included in Figure 4 (alongside the max probability method).

4. A complication of using high level functional information about kinases and substrates as the authors do here (GO terms, disease associations, pathways, etc.) is that the kinase-substrate relationships inferred can be indirect (involving multiple intermediate steps, e.g., “EGFR phosphorylates MAPK1”) rather than direct (substrate is directly phosphorylated by the kinase, e.g., “MAP2K1 phosphorylates MAPK1”). On the other hand, methods that use only “local” sequence information from the kinase and substrate would predict only direct kinase-substrate relationships. Arguably, the KSFinder method could be learning to predict only direct K-S relationships by using only direct relationships in its supervised training data. Both direct and indirect phosphorylation information can be useful depending on the type of downstream analysis of interest.

The authors should address this explicitly by 1) highlighting the distinction between direct and indirect phosphorylation relationships; 2) clarifying whether their aim is to predict only direct K-S relationships or both direct and indirect, and how that is reflected in the training data; 3) if the goal is only direct, attempting to evaluate the proportion of predicted K-S relationships that are plausibly direct, e.g. by cross-referencing with protein-protein interaction information and spot-checking for common indirect phosphorylation associations e.g., in the EGFR/MAPK pathway; and 4) considering the effects direct vs. indirect relationships in their assessments with other methods (for example, the site-specific LinkPhinder might be expected to perform poorly on predicting indirect K-S relationships.

Minor points:

- Figure legends should identify which assesments they are associated with, e.g. Figure 3 shows the results for Assessments 3-6, and Figure 4 shows the results for Assessments 7 and 8.

Line 465: The authors note that of the 17 positive predictions they found literature evidence for, 3 were found in a manual study of HIPK3 and NUAK2. Because there was no comparable manual literature search performed for the negative examples, the sensitivity score reported should presumably be calculated using only the automated results (14 / 18 = 0.78).

Line 242: How many negative triples are there in the Negatome dataset? This would be helpful in interpreting the significance of the 16 pairs that overlapped with the positive set.

244-251: The description of the negative sampling is somewhat hard to follow. It would be help if the description of the algorithm included the significance/relevance of the approach, e.g. after line 249, “This procedure ensured that substrates had a balance of both positive and negative triples in the datset.”

Very minor points/suggestions:

line 47: “predicts the kinase-substrate link” -> “predicts kinase-substrate links”
77: “the reversible post-translational” -> “a reversible post-translational”
78: “and regulates many processes” -> “that regulates many processes”
97: “in-vitro” -> “in vitro” (italicized)
105: “employed” -> “used a”
106: “the latent association of the” -> “latent associations of“
pred
426: “clubbing” -> “concatenating” or “combining”?
508: “and neural network” -> “and a neural network”
538: “in an extracellular matrix organization” -> “in extracellular matrix organization”

Experimental design

See Major Points 2 and 3 above.

Validity of the findings

See Major Points 2 and 3 above.

Reviewer 2 ·

Basic reporting

The manuscript describes KSFinder, an approach to predicting kinase-substrate relationships based on knowledge graph embeddings. First, a knowledge graph is constructed which contains thousands of known kinase-substrate relations as well as a much larger number of other relations surrounding these. First, embeddings models are learned over the KG, then a classifier is trained using embeddings. Results are benchmarked at different stages of this process, and ultimately the paper examines two understudied kinases in more detail. It is important to highlight that both the data and the source code are made available publicly which is a valuable contribution.

One key recent reference missing from the paper is Johnson et al. (https://doi.org/10.1038/s41586-022-05575-3) which provides an atlas of phosphosite-specific kinase-substrate predictions for a large part of the human kinome. The relationship to this work could be discussed and potential comparison could be made.

Experimental design

Methodological questions:
• Knowledge graph embeddings: one conceptual question is, to what extent are the four KGE algorithms best suited for the task of kinase-substrate prediction? The kinase-substrate prediction task is inherently many-to-many, since a single kinase often phosphorylates several downstream substrates, and conversely, a substrate often has multiple kinases phosphorylating it. For example, TransE, while being computationally efficient, inherently cannot model one-to-many, many-to-one, and many-to-many relationships. See e.g,:
Antoine Bordes, Nicolas Usunier, Alberto Garcia-Duran, Jason Weston, and Oksana Yakhnenko. 2013a. Irreflexive and hierarchical relations as translations. arXiv preprint arXiv:1304.7158.
Antoine Bordes, Nicolas Usunier, Alberto Garcia-Duran, Jason Weston, and Oksana Yakhnenko. 2013b. Translating embeddings for modeling multi-relational data. In Advances in Neural Information Processing Systems, pages 2787–2795.
A much broader set of models is available with the software PyKeen (https://github.com/pykeen/pykeen#models) which the authors could choose from if there are ones better suited for this task.
• Closed world assumption: negatives are generated with respect to the initial knowledge graph with any generated negative that is already in the KG being excluded. However, predictions of the model are then evaluated using a broader definition of positive triples where text mining and manual literature search confirms correct predictions from the model. This would imply that the negative examples generated for training could also actually have been positive with respect to existing literature. Can the authors comment on this issue?
• The final set of results involves enrichment analysis for two understudied kinases: HIPK3 and CAMKK1. The approach followed here is to first predict substrates for each kinase and then perform biological process enrichment over the set of substrates. Given that the KG itself also contains biological process terms and associations, this approach appears potentially circular. Given the goal of predicting functions of understudied kinases, could one make that prediction directly based on the KG rather than first learning a substrate prediction model, predicting substrates, and then doing enrichment over the substrates?

Validity of the findings

My main concern with the paper is the seemingly large number of false positive predictions that the model makes:
• Line 457 of the Results: “We found literature evidence supporting 17 of our 36,055 positive predictions involving the 68 understudied kinases”. The absolute number of positive predictions is very high: 36,055 positive predictions for 68 understudied kinases constitutes about 530 positive predictions per kinase. That number is one or two orders of magnitude higher than the number of substrates for typical human kinases which implies that most of these predicted positives would have to be false. Further, finding 17 out of 36,055 positive predictions in literature seems very low and the authors should provide better interpretation of what this means in terms of the quality of the model.
• Line 476 of the Results: “Of the proteins predicted to be phosphorylated by HIPK3, 230 scored a probability value greater than 0.9”. This is again an unrealistically high number in terms of the underlying biology, implying that the model likely predicts mostly false positives.
• Line 531-535 of the Discussion states that “we […] reported a sensitivity score of 0.81 with 17 true positives and 4 false negatives. The high precision of positive predictions indicates that KSFinder can be used to reliably generate predicted substrates”. The authors need to provide more proof or explain better how “high precision of positive predictions” is true given that the 17 true positives were picked from 36,055 overall positives. Even though it’s difficult to determine what proportion of the 36,055-17=36,038 positives are true or false positives, the fact that there are on average 530 positive predictions per kinase (as argued above) suggests that most of these are likely to be false positives.

Additional comments

Minor comments:
• Line 161: “20 different relationships” should probably be “20 different relation types”
• I found links to the source code and data along with the submission but these could be mentioned more prominently in the manuscript text as well.
• Figure 3: improve caption and legends to explain “different datasets”
• Figure 2: y-axis don’t start at zero, which makes small differences look larger than they are.
• The section on the KS-negative dataset (starting like 234) is somewhat hard to follow. It might be useful to have a figure/supplementary figure showing the approach.

Reviewer 3 ·

Basic reporting

Acceptable.

Experimental design

Anandakrishnan et al. employ a Knowledge Graph (KG) embedding approach to predict kinase-substrate interaction in this study. They report on a new method, KSFinder, that overcomes some of the limitations of previous approaches by generating new negative datasets. The authors also perform enrichment analysis to identify biological processes regulated by the predicted kinase-substrate interactions. Overall, the authors tackle an important question using a strategic negative dataset generation approach and test the performance of KSFinder on various datasets, including balanced and unbalanced ones. However, despite these advances, the current work has some limitations that need to be addressed before publication.

Major comments:

1. The novelty of KSFinder is limited. The KG embedding is generated using existing models like TransE, HolE, DistMult, and ComplEx that are pretty old (published nearly seven years ago) and not specially designed for link prediction tasks. The authors should benchmark against more recent methods.

2. The authors should make the code, testing, and training datasets available for review and replication of results by the scientific community.

3. The authors do not address the possibility of overfitting, as the usual random train test split would result in inflated scores.

4. The rationale for picking 17 out of 36,055 predictions is unclear. It comes across as cherry-picking. What is the false positive rate for predictions?

5. Details of how KSFinder generates probability scores should be provided in the methods and discussed in the results while comparing with other methods

Minor:

1. Under the section in Materials & Methods listed as "supervised learning under MLP classifier," I would suggest changing the wording. ComplEx is cited as State of the Art. However, the family of models ComplEX belongs to is still being actively improved. ComplEx is unable to capture the notion of composition (Comp. def.: r1(X, Y ) ∧ r2(Y, Z) ⇔ r3(X, Z) and Comp. def.: r1(X, Y ) ∧ r2(Y, Z) ⇔ r3(X, Z)). It cannot capture intersections Intersection: r1(X, Y ) ∧ r2(X, Y ) ⇒ r3(X, Y ) whereas models such as ExpressiveE (2023) can.

2. Can the authors clarify if they confirmed that subcellular location was different for the negative dataset chosen through cosine similarity of entries within the embedding space? It could be more apparent if the authors demonstrated that they were from separate locations- a supplemental figure depicting this analysis should be included.

Validity of the findings

Methods lack novelty, but this is not a major concern, given the focus of the paper in addressing a biological question, i.e, predicting kinase-substrate interactions

Additional comments

Acceptable for publication after major revision

---

## Round 0.2 · accepted · Accept

Thank you for your patience during this process and congratulations again.

Reviewer 3 ·

Basic reporting

The authors have satisfactorily addressed my comments and suggestions

Experimental design

no comment

Validity of the findings

no comment